# The Clinical Trial Landscape for Melanoma Therapies

**DOI:** 10.3390/jcm8030368

**Published:** 2019-03-15

**Authors:** Sonia Wróbel, Małgorzata Przybyło, Ewa Stępień

**Affiliations:** 1Department of Medical Physics, Marian Smoluchowski Institute of Physics, Jagiellonian University, 30-348 Krakow, Poland; soniawrobel1@gmail.com; 2Institute of Zoology and Biomedical Research, Faculty of Biology, Jagiellonian University, 30-387 Krakow, Poland; malgorzata.przybylo@uj.edu.pl

**Keywords:** melanoma, clinical trials, review, melanoma therapy, melanoma drug

## Abstract

(1) Despite many years of research, melanoma still remains a big challenge for modern medicine. The purpose of this article is to review publicly available clinical trials to find trends regarding the number of trials, their location, and interventions including the most frequently studied drugs and their combinations. (2) We surveyed clinical trials registered in the International Clinical Trials Registry Platform (ICTRP), one of the largest databases on clinical trials. The search was performed on 30 November 2018 using the term “melanoma”. Data have been supplemented with the information obtained from publicly available data repositories including PubMed, World Health Organization, National Cancer Institute, Centers for Disease Control and Prevention, European Cancer Information System, and many others to bring the historical context of this study. (3) Among the total of 2563 clinical trials included in the analysis, most have been registered in the USA (1487), which is 58% of the total. The most commonly studied drug in clinical trials was ipilimumab, described as applied intervention in 251 trials. (4) An increase in the number of melanoma clinical trials using immunomodulating monoclonal antibody therapies, small molecule-targeted therapies (inhibitors of BRAF, MEK, CDK4/6), and combination therapies is recognized. This illustrates the tendency towards precision medicine.

## 1. Introduction

The number of melanoma clinical tests is increasing every year, especially for patients with advanced stages of cancer. However, it is difficult to find statistical and geographical data in the literature showing trends in this area of clinical research. In this study, we aim to analyze melanoma clinical trials registered in the International Clinical Trials Registry Platform (ICTRP) [1]. The analyzed data allow us to catalog Food and Drug Administration (FDA) approved melanoma drugs, examine the number and locations of clinical trials registered each year, and identify interventions most frequently used in clinical trials.

Invasive melanoma remains the main cause of skin cancer deaths, despite being only about 1% of all skin cancer cases [2]. Hope for improvement is the dynamic development of new targeted therapies, immunotherapy, and combined therapies. Unfortunately, the effects of treatment are not always durable [3]. There is also a large heterogeneity of responses to the treatment used. Therefore, it is necessary to conduct clinical trials to check the effectiveness of new therapies, drugs combinations, or different dosages.

### 1.1. Melanoma Incidence and Mortality Rates

Worldwide, the incidence of cutaneous melanoma has risen dramatically. It is estimated that about 1.7% (232,100 cases) of all newly diagnosed primary malignant tumors as well as 0.7% (55,500 deaths) of all cancer deaths are caused by cutaneous melanoma annually [4]. The most cases of melanoma per 100,000 people are diagnosed in the United States of America and in Europe. According to an American Cancer Society report, in 2018 in the United States, 91,270 new cases of melanoma (55,150 in males and 36,120 in females) and 9320 deaths due to melanoma (5990 males and 3330 females) are estimated [2]. The risk of cancer averages 2.1%, which means that one in 50–60 Americans will develop melanoma in their lifetime [5]. The US Center for Disease Control and Prevention (CDC) published that the number of newly diagnosed cases in the US will reach 112,000 annually by 2030 [6]. United States Cancer Statistics (USCS) lists melanoma of the skin as the top sixth cancer by rates of new cancer cases [7]. Melanomas rates are reported to be different among various states. In 2015, most new cases occurred in Utah (40.5 per 100,000 people), Vermont (35.8 new cases per 100,000 people), Minnesota (31.6 new cases per 100,000 people), and New Hampshire (31.6 new cases per 100,000 people) [7]. Based on the European Cancer Information System (ECIS), in 2018, estimated incidence of skin melanoma in European Union 28 countries (EU28) is about 120,505 new cases (60,524 in males and 59,981 in females) [8]. Including countries belonging to the European Free Trade Association (EFTA), new cases will be about 126,390 (63,663 in males and 62,727 in females) in total (EU28 + EFTA) [8]. Melanomas incidence and mortality rates in Europe are very diverse depending on the country. Most new cases per 100,000 people are estimated in Norway (55.1), Netherlands (45.9), Denmark (45.2), and Sweden (43.8). The lesser numbers are estimated in Poland (10.3), Cyprus (8.4), Bulgaria (7.1), and Romania (5.7), respectively [8].

Because of these disturbing statistical data, it is necessary to conduct clinical trials for new therapeutic methods, especially for advanced stages of melanoma (only 20% five-year relative survival rate) [9]. In addition to treatment, rapid and effective diagnostics, including the search for new biomarkers, are also very important [10]. The aim of this article is to analyze publicly available melanoma clinical trials data to describe trends regarding research timelines, locations, and studied interventions.

### 1.2. Risk Factors and Genetic Alterations in Melanoma

Melanoma is considered a multifactorial disease in which both genetic susceptibility and environmental exposure play important roles. Established environmental risk factors for developing melanoma are (i) excessive, intermittent exposures to solar ultraviolet radiation and subsequent sunburns, and (ii) the use of UVA-emitting indoor tanning beds (especially when the use started before the age of 35 years) [11,12]. Cumulative UV radiation exposure is thought to be responsible for 60–70% of cutaneous malignant melanoma cases. UVB induces direct DNA damage in the form of photoproducts, including cyclobutane pyrimidine dimers (CPDs) and 6-4 photoproducts (6-4 PPs), accumulation of which could lead to DNA strand breaks. UVB also increases infiltration of macrophages and neutrophils into the skin [13,14] (Figure 1). The inflammatory response created by these cells promotes angiogenesis as well as melanoma cell survival, invasion, and angiogenesis. UVA is known to directly induce oxidative stress in melanocytes through the production of free radicals during the biochemical interaction of UVA with melanin, which causes the damage to DNA, lipids, and proteins [14].

The most important host risk factors for developing melanoma are (i) the presence and the number of melanocytic nevi and their appearance and involution throughout the life [15,16], (ii) a family history of cutaneous melanoma [16], (iii) multiple primary melanomas in a given individual [17], (iv) certain phenotypic characteristics such as red hair, fair skin, light eyes, sun sensitivity, an inability to tan, the familial atypical mole/melanoma (FAMM) syndrome [18], and (v) a high socioeconomic status [19].

Melanoma is a complex disease involving numerous genes and displaying the highest rates of mutation among all cancers [20]. The risk for developing melanoma is associated with both familial mutations and somatic mutations, but genetic predisposition is responsible for only a small number of cases. Susceptibility of some families is predominantly caused by a mutation in the germline *CDKN2A* gene and less frequently due to mutations in *CDK4* and *Rb1* genes [21,22,23]. The two most well characterized mutations commonly found in melanomas are *BRAF^V600E^* (~40%) and *NRAS^Q61^* (~20%) [14,22]. These mutations also occur with high frequencies in benign *nevi*. *BRAF^V600E^* mutation alone is not sufficient to drive malignant transformation of melanocytes. Driver mutations that promote melanoma progression include: the TElomerase Reverse-Transcriptase (*TERT*) promoter (progression into intermediate lesions and melanoma in situ), *CDKN2A* (gain of invasive potential), Phosphatase-and-TEnsin homologue (*PTEN*), tumor-protein p53 (*TP53*) (metastatic melanoma progression), Phosphatidylinositol-3,4,5-Trisphosphate Dependent Rac Exchange Factor 2 (*PREX2)* and its small GTPase substrate, (*RAC1)* (associated with melanocyte proliferation), and neurofibromin 1 (*NF1*) (typical for chronically sun-exposed skin or in older individuals). All of them show a high mutation burden, are wild-type for BRAF and NRAS, and are associated with dysregulation of the RAS/MAPK pathway [22,24,25,26,27,28,29]. These genetic alterations affect intracellular signaling pathways like the Mitogen-Activated-Protein-Kinase (*MAPK*) pathway, the PhoshoInositide-3-Kinase (*PI3K*), protein-kinase-B (*AKT*), *PTEN*, and mammalian-Target-Of Rapamycin (*mTOR*) pathways. Minor percentages of melanomas have activating mutations in the *KIT* gene, most common in mucosal melanomas derived from the genital regions or mutations in G Protein Subunit Alpha 11 (*GNA11*) or G Protein Subunit Alpha q (*GNAQ*) genes in uveal melanomas [30,31].

Based on the pattern of the most prevalent mutations, four different genetic melanoma subtypes were delineated: (i) *BRAF* mutant melanomas (~50% of melanomas), (ii) *RAS* (*N*,*H*,*K*) mutant melanomas (~25% of melanomas), (iii) *NF1* mutant melanomas (~15% of melanomas), and (iv) *BRAF/NRAR/NF1*-(triple) wild-type melanomas (~10% of melanomas) [32]. Molecular and epidemiological data support the hypothesis that there are two distinct biological pathways that might trigger melanoma development: (i) *BRAF*-associated *nevus* prone pathway, which is initiated by early sun exposure and promoted by an intermittent sun exposure or possible host factors, characterized by young age at diagnosis and absence of chronic sun damage on the skin, and concerns mainly melanomas on the trunk and superficial spreading melanomas; and (ii) a chronic sun exposure pathway in sun sensitive people who progressively accumulate sun exposure to the sites of future melanomas—this pathway is characterized by *NRAS* mutations without any associations with *nevus* count [33] (Figure 2).

## 2. Methodology of Data Search

In total, 2563 clinical trials registered in the ICTRP database were analyzed [1]. ICTRP contains data from the largest authorities registering clinical tests. The database is constantly updated and includes data from Australian New Zealand Clinical Trials Registry, Chinese Clinical Trial Registry, ClinicalTrials.gov, EU Clinical Trials Register (EU-CTR), International Standard Randomised Controlled Trial Number Registry (ISRCTN), The Netherlands National Trial Register, Brazilian Clinical Trials Registry (ReBec), Clinical Trials Registry—India, Clinical Research Information Service—Republic of Korea, Cuban Public Registry of Clinical Trials, German Clinical Trials Register, Iranian Registry of Clinical Trials, Japan Primary Registries Network, Pan African Clinical Trial Registry, Sri Lanka Clinical Trials Registry, Thai Clinical Trials Registry (TCTR), and Peruvian Clinical Trials Registry (REPEC). The search was performed on 30 November 2018 using the term “melanoma”. Trials in which melanoma was mentioned as a condition the test referred to were included in analysis without time or geographical restrictions. Historical information about clinical trial outcomes and details was obtained from PubMed [34], FDA [35], World Health Organization (WHO) [36], American Cancer Society (ACS) [37], National Cancer Institute (NCI) [38], United States Cancer Statistics (USCS) [7], CDC, ECIS [8], and Cancer Australia [39].

The dataset obtained was analyzed with the use of a proprietary script written in Python. The results obtained for the most common interventions were used to create the bubble chart. For each intervention, “unique” occurrences were counted, e.g., if a drug appeared in one clinical trial several times, for example, in combinations with different drugs, it was counted as one occurrence. The obtained results reflect the number of “unique” clinical tests in which the medicine was given as the intervention used. This approach allowed us to avoid “duplicates”.

## 3. Historical Melanoma Clinical Trials Breakthroughs—FDA Approval Overview

The first clinical trial on melanoma registered in ICTRP started in 1971 in Argentina. In 1975, the FDA approved dacarbazine for advanced, metastatic melanoma considering the progress in melanoma therapy to gradual improvement and development (Figure 3, Table 1). Later on, in 1984, Edward Creagan and John M. Kirkwood started the first trials of recombinant interferon for resectable melanoma [40]. In the same year, Lloyd J. Old, Herbert F. Oettgen, and Alexander Knuth demonstrated in the foremost clinical experiments that interleukin-2 (IL-2)-dependent T cells could be trained to recognize and attack malignant melanoma [41]. The year after this discovery (1985), Steven A. Rosenberg and his team reported the mediation of cancer regression in humans by an IL-2-based immunotherapy [42]. In 1987, Rosenberg and colleagues treated the first patient with metastatic melanoma with immunotherapy based on autologous tumor-infiltrating lymphocytes and IL-2 [43]. Until the 1990s, only a few clinical tests had been registered (Figure 4).

The 1990s were full of groundbreaking discoveries in the field of melanoma treatment. It started in 1991 when the Radiation Therapy Oncology Group published the first randomized trial using radiotherapy to treat melanoma (RTOG 83-05) [44,45]. In 1993, Giorgio Parmiani and his team described human melanoma antigens recognized by patient T cells [46]. One year later (1994), Alan Houghton, Carl F. Nathan, and colleagues demonstrated the pilot trial of treatment with monoclonal antibodies in patients with metastatic melanoma [47]. Bijay Mukherji’s team reported the first dendritic cell vaccination for human melanoma in 1995 [48]. At the end of 1995, Marie Marchand, Thierry Boon, and their colleagues described the first clinical regressions of melanoma metastases after vaccination with a MAGE-A3 peptide [49].

In 1996, the FDA approved high-dose interferon in melanoma after a multicenter, randomized, controlled phase III study from the Eastern Cooperative Oncology Group directed by John M. Kirkwood. Reports showed “the first significant improvement of both survival and relapse-free interval for melanoma patients treated with high-dose interferon alfa-2b (E1684)” [50].

At the beginning of the 21st century, in 2001, The Cancer Research Institute and the Ludwig Institute for Cancer Research established the Cancer Vaccine Collaborative (CVC), including a melanoma cancer vaccine program. In 2002, the first clinical trial dedicated to “emerging class of immunomodulatory antibodies” (monoclonal antibodies to induce CTLA-4 blockade) in melanoma patients was published [51]. In the same year, Helen Davies described a high frequency of *BRAF* mutation in melanoma, which brought new insight in melanoma etiology [52].

The group of scientists Christophe Lurquin, Thierry Boon, and Pierre Coulie reported post-vaccination melanoma regression in 2005 [53,54]. One year after this discovery, Steven A. Rosenberg and colleagues announced melanoma regression in patients after transfer of genetically engineered lymphocytes (bulk T cells transduced with T cell receptor genes) [55]. In 2006, Rosenberg’s team published a paper about this new adoptive T cell technology for melanoma treatment [56].

The first neoadjuvant trial of interferon (UPCI 00-008) finished in 2007 with spectacular immunological effects. In 2011, the FDA approved three drugs: pegylated interferon (Sylatron™), vemurafenib (Zelboraf^®^, PLX4032), and ipilimumab for melanoma treatment. Ipilimumab (Yervoy^®^) was the first drug in history that provided a significant increase in the survival rate for patients with advanced melanoma in randomized phase III clinical trials [57]. At the end of 2011, NCI appointed 27 members for the Cancer Immunotherapy Trials Network.

The phase I clinical trial with Bristol-Myers Squibb’s anti-Programmed cell Death protein 1 (PD-1) antibody showed, in 2012, very promising results with tumors regression in 28% of melanoma patients. In 2013, Padmanee Sharma and colleagues described ICOS+ CD4 T cells biomarker that could be used for identification of melanoma patients who are more likely to respond to ipilimumab [58]. In 2013, the FDA approved dabrafenib (Tafinlar^®^) and trametinib (Mekinist^®^), and the Japan drug administration approved nivolumab under the name Opdivo^®^. Next year (2014), the FDA approved pembrolizumab (Keytruda^®^) and nivolumab for unresectable or metastatic melanoma. Pembrolizumab was the first and nivolumab was the second *PD-1* inhibitor applied for melanoma therapy in the US.

At the end of the 2014, Mario Sznol and his team reported that in the phase Ib of clinical trials using an intervention combination of nivolumab and ipilimumab in advanced melanoma patients (unresectable or metastatic melanoma), they obtained one- and two-year survival rates of 94% and 88% [59]. After this discovery, the FDA approved this combination. Nowadays, combined therapies are a great hope. The FDA approved a combination of dabrafenib and trametinib in 2014 and a combination of combimetinib (Cotellic^®^) and vemurafenib in 2015, both for patients with unresectable or metastatic melanoma with *BRAF^V600E^* or *BRAF^V600K^* mutations (Table 2). In 2017, the FDA approved a combination of dabrafenib and trametinib for those patients and nivolumab for the adjuvant treatment. Both therapies are dedicated to patients with involvement of lymph nodes or who have complete resection [36]. In 2018, the FDA approved a combination of encorafenib (Braftovi^®^) and binimetinib (Mektovi^®^) for unresectable or metastatic melanoma with *BRAF^V600E^* or *BRAF^V600K^* mutations [60].

In 2015, the FDA announced the first in class oncolytic virus therapy for advanced melanoma called talimogene laherparepvec (Imlygic™, Amgen). One year later (2016), the FDA modified the dosage for nivolumab [60].

The number of registered studies was growing systematically within the period of 1971–2018 (Figure 4a). The largest number of clinical trials (198) was registered in 2015. The most frequently mentioned melanoma interventions shows pembrolizumab and nivolumab on the top of 2017 and 2018 (Figure 4b).

## 4. Melanoma Trials Location around the World

Until now, melanoma clinical trials were registered in 72 different countries (Figure 5). Of the total 2563, all European countries were reported as locations for 869 clinical trials, North America for 1529, Asia for 291, Australia and Oceania for 275, South America for 95, and Africa for 31, respectively. The largest number was registered in the United States (1487), 58% of the total number of melanoma clinical trials. Among the entire number, 1006 trials (39%) were registered as multicenter studies, 1457 as single center, and 100 with non-location information. Unfortunately, no data are available for almost the whole of Africa, except the Republic of South Africa with 30 registered clinical trials and Egypt with two clinical trials registered.

## 5. Melanoma Clinical Interventions

Eleven classes of intervention are given in the database: behavioral (e.g., questionnaires, survey, self-examination of the skin mole mapping diagram), biological (e.g., vaccines, interleukines), combination product, device (e.g., positron emission tomography, electrical impedance spectrometer, ultrasonography, magnetic resonance imaging (MRI), computed tomography (CT)), diagnostic test (e.g., positron emission tomography/computed tomography (PET/CT) scan, clinical decision support), dietary supplement (e.g., genistein, fish oil supplement, calcitriol, citicoline), drug, genetic (e.g., gene expression analysis, protein expression analysis, DNA methylation analysis), procedure (e.g., biopsy, surgery, blood sampling), radiation (e.g., radiotherapy, stereotactic radiosurgery, stereotactic body radiation therapy) and other (e.g., laboratory biomarker analysis, immunohistochemistry staining method, educational intervention, photographs). Interventions such as “behavioral” or “dietary supplement” have usually not been used in isolation but as an adjunct intervention with a specific therapy or diagnostic. To observe the development of therapies and biomarkers fields, “biological”, “combination product”, “drug”, “genetic”, and “other” categories were selected for detailed analysis.

The most often used intervention in melanoma clinical trials was ipilimumab (immunotherapy checkpoint inhibitor, targeting CTLA-4), described as the applied intervention in 251 trials (Figure 6). The next five more often applied interventions were: aldesleukin (IL-2) used in 166 clinical trials, pembrolizumab (immunotherapy checkpoint inhibitor, targeting PD-1) used in 131 clinical trials, nivolumab (immunotherapy, checkpoint inhibitor, targeting PD-1) used in 124 clinical trials, radiotherapy used in 144 clinical trials, and cyclophosphamide (chemotherapy) used in 103 clinical trials.

The most frequently exchanged drug monotherapies were: chemotherapies (cyclophosphamide, temozolomide); immunomodulating monoclonal antibodies therapies (anti-PD-1/PD-L1, anti-CTLA-4); and small molecule-targeted therapies (BRAF, MEK, or CDK4/6 inhibitors). In recent years (2017-2018), the most frequently tested drugs were pembrolizumab, nivolumab, ipilimumab, and atezolizumab as immunotherapies; trametinib, dabrafenib, cobimetinib, and vemurafenib as targeted therapies; and others were cyclophosphamide (chemotherapy), talimogene (oncologic virus), and IL-2 (Figure 2b).

Combined drug therapies have been studied more frequently in recent years due to achieving very promising outcomes, especially in cases of advanced melanoma. The most promising ones are:immunotherapy combinations: anti-PD-1 plus anti-CTLA-4, anti-PD-1 plus IDO inhibitor, anti-PD-1 plus anti-LAG-3, anti-PD-1/anti-CTLA-4 plus HDAC inhibitor, anti-PD-1/anti-CTLA-4 plus TLF-9 agonist, anti-PD-1 plus anti-GITR, anti-PD-1 plus pegylated IL-2 (NKTR-214)targeted therapy combinations: MEK inhibitor plus CDK4/6 inhibitor, triplet combination of BRAF plus MEK inhibitor plus CDK4/6 inhibitorcombined immunotherapy and targeted therapy: triplet combination of BRAF/MEK inhibition with anti-PD-1combination of immunotherapy and cell therapy

Patients who have cancer progression after immunotherapy can receive good results by implementation of adoptive cell transfer—the infusion of large numbers of activated autologous lymphocytes. The effect of intestinal microflora on the response to immunotherapy is also tested. In 2018, a clinical trial was started with fecal microbiota transplant (FMT) combined with pembrolizumab in melanoma patients.

## 6. Discussion

The ICTRP is a large clinical trials database, but it is not the only source of information about clinical trials. This is the reason why this review has limitations—there may be clinical trials that have not been recorded in this database. Not every clinical trial is documented in exactly the same way. Differences or inaccuracies in the descriptions may have caused omission in the analysis. In order to improve data reliability and completeness, information was supplemented from such sources as PubMed, the FDA, the WHO, and others.

In 1971, the first clinical trial year, US president Richard Nixon signed the National Cancer Act [62]. This document not only established not only the National Cancer Institute we know today but also a model for building public–private cooperation around a worldwide network of research laboratories and oncology centers. It was a US commitment to what President Richard Nixon called a “War on Cancer”. Research on new cancer therapies received very large funding, which also resulted in the increase in the number of newly started cancer clinical trials.

In 1997, members of the Pharmaceutical Research and Manufacturers Association (PhRMA) leading biopharmaceutical researchers and biotechnology companies declared spending $1.4 billion on cancer research. This trend can also be seen in the National Cancer Institute budget. In 1992, they spent $24.8 million on melanoma research. This amount constantly grew to reach, in 1997, $54.8 million to be used exclusively for melanoma [63]. After this year, a dynamic increase in the number of clinical trials in this area can be noticed (Figure 2). National Cancer Institute Research Funding for melanoma reached $132.8 million in 2015 [8].

Most of our current knowledge about melanoma comes from the research of patients from the United States and European populations. It is noteworthy that some subtypes of melanoma, which most often develop in other populations, are not associated with high exposure to UV. This suggests a different etiology of the disease. Further examination of these subtypes is necessary to identify their risk factors (including genetic factors) and to adapt therapies to benefit all ethnic groups [64].

In recent years, a significant increase in clinical trials has been observed with the use of immunotherapies and targeted therapies for melanoma. According to the latest report of the Cancer Research Institute (CRI), 2250 clinical trials are currently underway in the world in the evaluation of efficacy of checkpoint inhibitors of anti-PD-1/PD-L1 [65]. It means an increase of 748 studies over the last year compared to last year, a 33% increase in the number of active clinical trials, and a 31% increase in new therapeutic goals tested in combination with checkpoint inhibitors. This tendency in the entire area of melanoma drugs is similar and illustrates the new direction towards precision medicine analyzing the molecular characterization of the patient’s tumor.

## Figures and Tables

**Figure 1 jcm-08-00368-f001:**
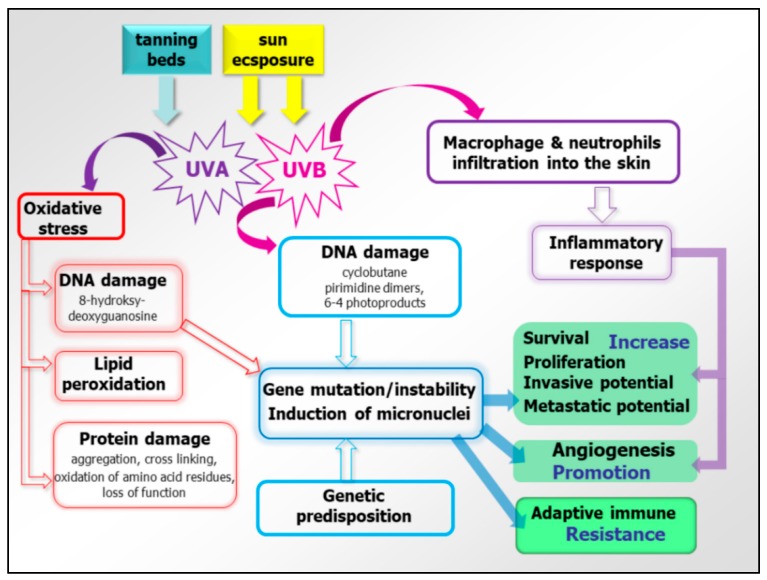
Molecular background of UV radiation and oxidative stress for the melanoma cause. Both sun exposure and tanning beds are sources of UV. Among them, UVA is mostly responsible for oxidative stress, causing DNA and protein damage as well as lipid peroxidation. DNA damage leads to gene instability and mutagenesis, impacting the main cellular processes involved in cancerogenesis, cancer growth, metastasis, and adaptive immune resistance. Likewise, UVB modulate those processes by the activation of inflammatory response.

**Figure 2 jcm-08-00368-f002:**
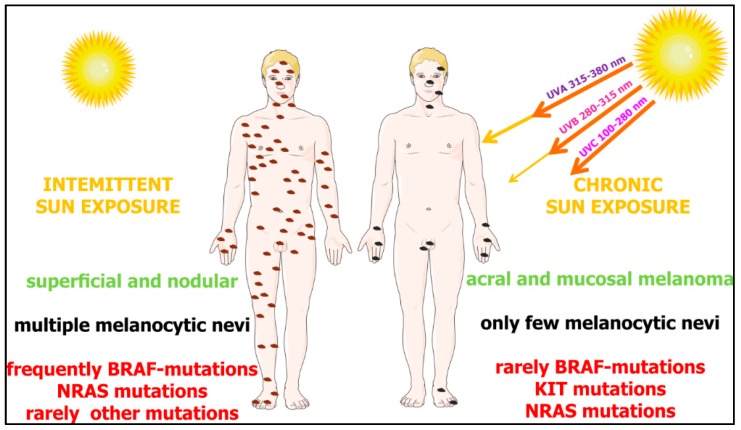
Differential dependence of melanoma subgroups on sun exposure and types of gene mutations. The interplay of genetic and environmental factors on melanoma development. Up to 50% of melanomas derived from the skin without chronic sun damage (intermittently exposed to UV) contain *BRAF* mutations. The second frequent mutations are present in the *NRAS* gene (around 25%). In melanomas derived from chronic sun exposure, *NRAS* and *KIT* mutations are more common (~15% and ~20%) than *BRAF* (~10%). According to Janina Staub (2012) with some modifications [30,31].

**Figure 3 jcm-08-00368-f003:**
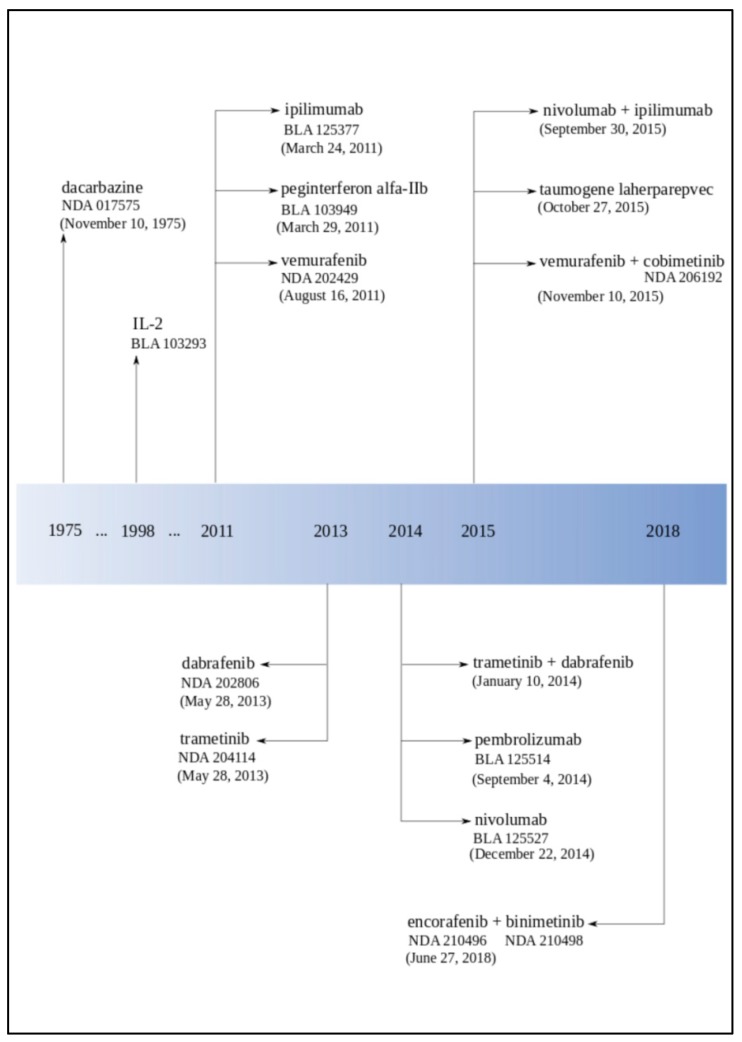
The Food and Drug Administration (FDA) approvals for melanoma drugs timeline [35].

**Figure 4 jcm-08-00368-f004:**
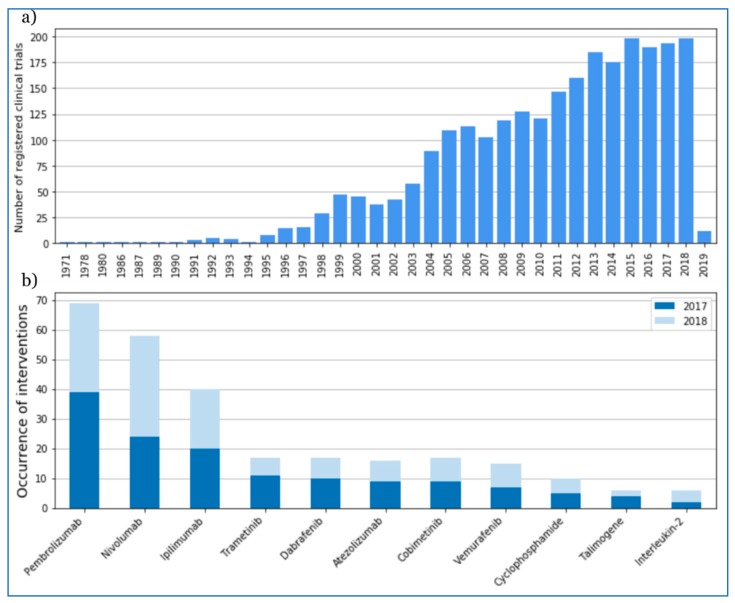
Number of trials starting each year from 1971 to 2019 based on International Clinical Trials Registry Platform (ICTRP) data (**a**). Most frequently mentioned melanoma interventions in the ICTRP database in 2017 and 2018 (**b**).

**Figure 5 jcm-08-00368-f005:**
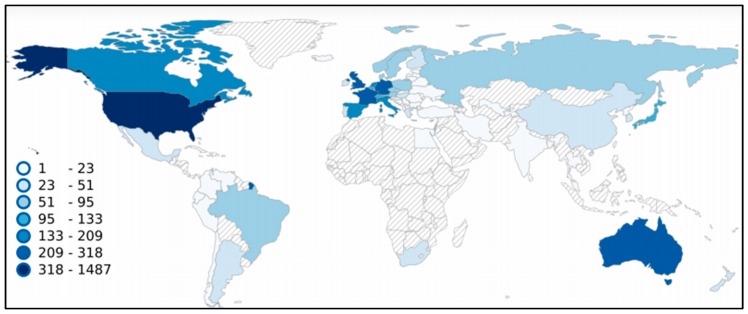
Locations of melanoma clinical trials by country. The map was generated based on data from the ICTRP [1]. Our search was performed on 30 November 2018 using the term “melanoma”. All results with location data are presented as a heatmap.

**Figure 6 jcm-08-00368-f006:**
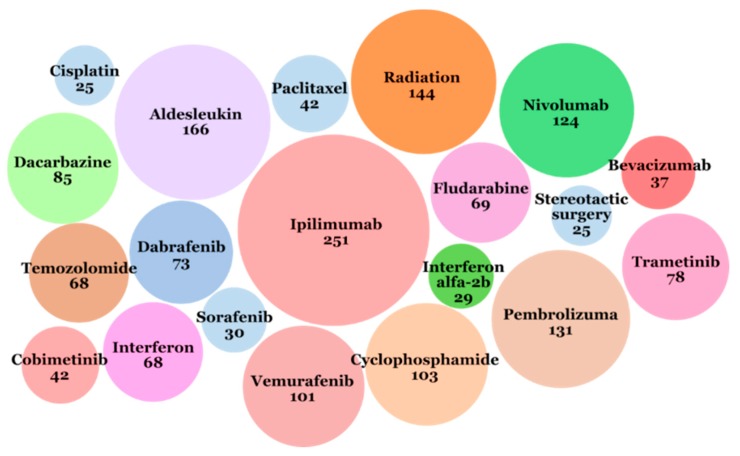
Twenty interventions that most often appeared in the ICTRP database from 1971 to 2018. The number under the name of intervention means the number of clinical trials in which the intervention was applied.

**Table 1 jcm-08-00368-t001:** FDA approved melanoma drugs [60,61].

Drug Name	Brand	Type	Biological Classification	Target	First FDA Approval
Monotherapies					
dacarbazine	DTIC-Dome^®^ Bayer	small-molecule	antineoplastic agent	DNA	November 1975
IL-2 (aldesleukin)	Proleukin^®^ Nestle	protein	interleukin	agonist/modulator of IL-2 receptor subunit alpha/beta, agonist of cytokine receptor common subunit gamma	January 1998
ipilimumab	Yervoy^®^ Bristol Mayer Squibb	antibody immune therapy	monoclonal antibody (mAb)	antagonist of CTLA-4	March 2011
peginterferon alfa-IIb	Peginton^®^ Merck&Co	small-molecule	interferons	agonist of interferon alpha/beta receptor 1 and 2	March 2011
vemurafenib	Zelboraf^®^ Roche/Genetech	small-molecule signaling antagonist	kinase inhibitor	mutant BRAF	August 2011
dabrafenib	Tafinlar^®^ Novartis Pharmaceuticals Corp.	small-molecule signaling antagonist	kinase inhibitor	mutant BRAF	May 2013
trametinib	Mekinist^®^ Novartis Pharmaceuticals Corp.	small-molecule signaling antagonist	kinase inhibitor	MEK	May 2013
pembrolizumab	Keytruda^®^ Merck&Co	antibody-immune therapy	monoclonal antibody (mAb)	antagonist of PD-1	September 2014
nivolumab	Opdivo^®^ Bristol Mayer Squibb	antibody-immune therapy	monoclonal antibody (mAb)	antagonist of PD-1	December 2014
taumogene laherparepvec	Imlygic^®^ Amgen	oncologic virus	virus	DNA polymerase catalytic subunit, heparan sulfate	October 2015

**Table 2 jcm-08-00368-t002:** FDA approved melanoma drugs combination [60,61].

Drugs Combination	Brand	Type	First FDA Approval
trametinib (Mekinist^®^) + dabrafenib (Tafinlar^®^)	Novartis	small-molecule targeted therapy	January 2014
nivolumab (Opdivo^®^) + ipilimumab (Yervoy^®^)	Bristol Mayer Squibb	antibody-immune therapy	September 2015
vemurafenib (Zelboraf^®^) + cobimetinib (Cotellic^®^)	Roche/Genetech	small-molecule targeted therapy	November 2015
encorafenib (Braftovi^®^) + binimetinib (Mektovi^®^)	Array BioPharma	small-molecule targeted therapy	June 2018

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
