# Peer review of "The Clinical Trial Landscape for Melanoma Therapies"

_jcm, 2019, doi:10.3390/jcm8030368_

Reviewer 1 Report

Review report

-        Manuscript no.: jcm-444759

-        Title: The clinical trial landscape for melanoma therapies

-        Corresponding Author: Ewa Stępień

-        Considered for publication to Journal of Clinical Medicine (MDPI).

General evaluation

In this manuscript Authors review the current state-of-the-art in melanoma chemo- and immunotherapies. The Authors exploited the ICTRP Database, a publicly available repository storing all the data associated to all clinical trials worldwide.

The manuscript is grammatically well written but there are some key compulsory major/minor concerns to be solved, described below. In general, Authors need to accurately check and correct the number/values in figures 1,2,3, and 4 they obtained by interrogating the ICTRP database.

Major concerns

1) Figure 1: Please insert the FDA orange book (https://www.accessdata.fda.gov/scripts/cder/ob/index.cfm) IDs, or other equivalent IDs, in the figure legend. E.g., in the year 2011 it has been reported that vemurafenib therapy has been activated on august 25, 2011. The associated FDA orange book ID is NDA202429, and so on. This will enable readers to better identify each depicted therapy in the figure. Authors should also carefully check all the dates indicated for each therapy in the figure.

2) Authors should describe, in the figure 3 legend, how they generated the data shown in that figure. For example, whchc public database they interrogated? And what keywords they used to generate the heatmap?

3) Figure 4: The Authors are suggested to carefully check the numbers depicted for each drug in Figure 4. Eg.., if one do perform a search in the ICTRP database by using the keywords “melanoma” and “Nivolumab” between “jan 1, 1971” and “dec 1, 2018” and in “ALL” trials, the search provides 213 records, quite different from the value currently showed in the circle (124).

Furthermore, Authors may also wish to generate a second similar panel in which the extents of combination therapies are depicted.

4) Why Authors does not show the succeeded trials vs unsucceeded in terms of patients survival rate? For example? The 124 trials depicted for Nivolumab therapy in Figure 4 how many of those have been successful (patient still alive)? And how many unsuccessful (patient deceased)?

Minor concerns

1) Sentence in row 75-87, page 2: To me it appeared that the queue yielded 2829 rather than 2563.

2) Row 89, page 2: correct the erroneous term “dacarbazine” witht the correct one “dacarbazine”.

Author Response

To Reviewer 1

1) Figure 1: Please insert the FDA orange book (https://www.accessdata.fda.gov/scripts/cder/ob/index.cfm) IDs, or other equivalent IDs, in the figure legend. E.g., in the year 2011 it has been reported that vemurafenib therapy has been activated on august 25, 2011. The associated FDA orange book ID is NDA202429, and so on. This will enable readers to better identify each depicted therapy in the figure. Authors should also carefully check all the dates indicated for each therapy in the figure.

We improved the Figure 1 including FDA orange book IDs. Because we have added more 2 figures 9accordding to the suggestions of the Reviewer 2, the numerations has changed.

2) Authors should describe, in the figure 3 legend, how they generated the data shown in that figure. For example, which public database they interrogated? And what keywords they used to generate the heatmap?

Explained in the legend

3) Figure 4: The Authors are suggested to carefully check the numbers depicted for each drug in Figure 4. Eg.., if one do perform a search in the ICTRP database by using the keywords “melanoma” and “Nivolumab” between “jan 1, 1971” and “dec 1, 2018” and in “ALL” trials, the search provides 213 records, quite different from the value currently showed in the circle (124).

In fact, the number of clinical trials for nivolumab in ICTRP is 124 for those dates. The discrepancy from a simple “data search” and our analysis arises form the different approach. In our analysest, the dataset obtained form ICTRP was analyzed with the use of a proprietary script written in Python. For each intervention, "unique" occurrences were counted, i.e. if a drug appeared in one clinical trial several times, e.g. in combinations with different drugs, it was counted as one occurrence. The obtained result reflects the number of “unique” clinical tests in which the medicine was given as the intervention used. This approach allowed us to avoid "duplicates". The appropriate explanation has been added do thee text in lanes: 150-155

Furthermore, Authors may also wish to generate a second similar panel in which the extents of combination therapies are depicted.

Unfortunately, not all information on combination therapies is included in ICTR database. To avoid erroneous conclusions related to the automatic analysis of incomplete data, only the selected most promising types of combination therapies with examples are described: immunotherapy combinations, targeted therapy combinations, combined immunotherapy and targeted therapy, combination of immunotherapy and cell therapy. The authors were not able to use their approach to analyses the combination therapies from those repositories.

4) Why Authors does not show the succeeded trials vs unsucceeded in terms of patients survival rate? For example? The 124 trials depicted for Nivolumab therapy in Figure 4 how many of those have been successful (patient still alive)? And how many unsuccessful (patient deceased)?

Please read above mentioned explanation.

Minor concerns

1) Sentence in row 75-87, page 2: To me it appeared that the queue yielded 2829 rather than 2563.

The discrepancy goes from the approach with the use of our algorithm, which allows to exclude the duplicated records.

2) Row 89, page 2: correct the erroneous term “dacarbazine” witht the correct one “dacarbazine”.

Corrected

Reviewer 2 Report

The current work is technically sound and well presented. It is well structured and provide evidences for the proof of concept.

However, there are additional concerns:

1. Provide a background for melanoma such as the cause e.g. UV radiation, oxidative stress.  Also provide the genetic association (provide genes involved in melanoma development).

2. It would be nice to highlight the melanoma and the various molecular perturbations diagrammatically.

Author Response

1. Provide a background for melanoma such as the cause e.g. UV radiation, oxidative stress.  Also provide the genetic association (provide genes involved in melanoma development).

We added the appropriate text with the new Figure 1

2. It would be nice to highlight the melanoma and the various molecular perturbations diagrammatically.

New Figure 2 presents this molecular and environmental background of melanoma

Round  2

Reviewer 1 Report

Review report on Version 2

-        Manuscript no.: jcm-444759-v2

-        Title: The clinical trial landscape for melanoma therapies

-        Corresponding Author: Ewa Stępień

-        Considered for publication to Journal of Clinical Medicine (MDPI).

Decision: Minor revisions required.

General evaluation

In this manuscript Authors do review the current state-of-the-art in melanoma chemo- and immunotherapies. In this revised version (v2) the Authors included two more figures to underline the role of UV irradiation in melanoma resistance/susceptibility to therapies.

Overall, the Authors fully answered to the comments of this Reviewer. There are only some minor issues to be solved, listed below.

Minor concerns

1) Please redraw the figure 2 by removing the sentence “Differential ….. gene mutations” and by putting it into the legend. Please provide more more detail about this figure in the legend.

2) Please provide a more detailed legend on figure 1 (the figure explaining the role of UVA/UVB).

Author Response

The authors thank for comments. The corrections of the Figure 2 has been done and the legends to Figure 1 and 2 were revised and extended.  

Reviewer 2 Report

The authors have performed the necessary amendments in the revised version. Overall, the manuscript is well presented in the current form.

Author Response

The authors thank for comments. Minor revision has been done.